# Japanese Family Conditions Demonstrating Family Resilience: Directed Content Analysis Based on Literature and Family Interviews

**DOI:** 10.3390/nursrep15030096

**Published:** 2025-03-13

**Authors:** Naohiro Hohashi, Natsumi Kijima

**Affiliations:** Division of Family Health Care Nursing, Graduate School of Health Sciences, Kobe University, Kobe 654-0142, Japan; nkijimaa@gmail.com

**Keywords:** family resilience, family nursing, item development, directed content analysis, Concentric Sphere Family Environment Theory

## Abstract

**Background/Objectives:** When experiencing a variety of negative family events, families that, as a whole, have high family resilience maintain and improve family functioning. It is important, therefore, for nursing professionals to be able to assess families lacking in family resilience in order to prevent, reduce, or ameliorate family symptoms (such as family-perceived problems, issues, difficulties, or suffering). The purpose of this study was to clarify how family resilience works. **Methods:** Family resilience is defined based on the Concentric Sphere Family Environment Theory (CSFET) as when a family becomes aware of family symptoms on its own, and its power to autonomously and actively improve its own family functions. The contents of 22 family nursing cases from 16 qualitative studies on family resilience and the results of 28 semi-structured interviews with 28 parenting-age families in Japan were qualitatively categorized using the triangulation method, and then directed content analysis was conducted based on the CSFET. **Results:** A total of 157 labels, with a total of 23 categories and 47 subcategories, were found to relate to family resilience, such as “can utilize relatives”, “family members can communicate with members of other families”, “family members can share information”, “all family members can communicate with one another”, “all family members can cooperate with one another”, and “can share time with family”. **Conclusions:** From these, a variety of diverse aspects contributing to a family’s resilience, including family member interactions, entire family interactions, use of social resources, and religious and spiritual support, were indicated.

## 1. Introduction

Family nursing aims at maintaining and improving family functioning in the target family system unit, and it targets the family as a living organization. However, in today’s Japan, families are becoming more diverse, family ties are weakening, and family functioning has been declining, making it more difficult for families to solve problems on their own [1]. It is considered that rather than unilateral family intervention/support from others to solve problems, families are more likely to improve their self-care skills through the support of others and to continue to practice the solutions they themselves have devised, which are more likely to be put into practice [1]. Under these circumstances, the concept of “family resilience” has attracted much attention in family nursing.

Resilience is a construct that has garnered considerable attention from social scientists, policymakers, and public and private organizations. Resilience has long been emphasized as warranting additional research [2,3]. The origins of the concept of resilience stem from early psychiatric literature in which children who appeared to be invulnerable to adverse life situations were examined. Resilience, the ability to recover or cope successfully despite substantial adversity [4], has received significant attention from various domains [5]. Although many deprived or traumatized children were hindered from psychological growth or found themselves confined to a situation in which they were victimized, Rutter [6] noted that about half of the children exposed to high-risk circumstances and who overcame them were able to lead loving and productive lives and went on to raise their own children well. However, in the 1990s, research on people who had been at risk but whose problems did not manifest themselves was conducted, with an interrelationship between the individual and the environment observed and redefined as a dynamic and changeable process [7]. Research on resilience expanded to include various age ranges and disadvantaged states, such as poverty and violence [6,8], abuse [9], chronic illness [10,11], and catastrophic life events [12,13].

Resilience has been viewed as residing within the individual, with the family often dismissed as dysfunctional [14]. Since the late 1990s, however, attention has focused on families that have demonstrated strength and problem-solving skills in the face of a crisis. Walsh’s adaptation of the concept of resilience in the family led to the study of family resilience, which emerged from psychology and was used in a variety of domains [14]. Subsequently, McCubbin and McCubbin [15] have defined family resilience as “characteristics, dimensions, and properties of families which help families to be resilient to disruption in the face of change, and adaptive in the face of crisis situation”, and Masten and Barnes [16] defined family resilience as “the capacity of a family to handle a specific impending crisis”. The written meanings do not differ greatly depending on the researchers. This concept is also applied to family nursing and is useful in formulation of responses to one of the enduring mysteries of family dynamics: why some families continually live well together and respond positively to challenges, whereas others in similar circumstances do not cope well. Family resilience entails more than surviving a crisis but also offers the potential for growth out of adversity. By weathering a crisis together, a family can emerge more loving, stronger, and more resourceful in meeting future challenges [11,17].

To provide family nursing, it is first necessary to identify which families need family nursing intervention. One example of a family condition that requires family nursing is the state in which family resilience is lacking. In order to provide intervention to families experiencing difficulties due to a lack of family resilience, assessment criteria are needed to facilitate determining which families need intervention.

Hohashi [18], a family nursing researcher/practitioner, proposed the Concentric Sphere Family Environment Theory (CSFET), a middle-range family nursing theory that is widely used in family nursing, that holistically aims at the high-dimension family existence, approaching the family system unit from the aspect of chrono and spatial axes, focusing on the family environment that acts on family well-being (Figure 1). CSFET includes family resilience as a functional indicator, and family resilience is defined as “when a family becomes aware of family symptoms, its power to autonomously and actively improve its own family functions” [1]. In order to enable family assessments of the family environment and condition of family well-being upon which the CSFET is based, the Family/Family Environment Assessment Model (FFEAM) was developed and well organized to correspond with the system of practice [19]. Specifically, the FFEAM [19] consists of a “family observation/interview” containing seven tools and a “measurement test” containing seven tools. In this way, multiple tools were developed and introduced, which enabled even beginners to collect data with high efficiency and reliability, and enabled family assessments. However, FFEAM does not yet offer a scale to screen for the degree of family resilience.

In this study, we conduct directed content analysis of the literature and semi-structured interviews using the triangulation method in order to understand how family resilience works, which means family conditions, based on the framework of CSFET. This will enable the future development of a scale to assess the degree of family resilience.

## 2. Methods

### 2.1. Definition of Terms

We define “the circumstances in which the family demonstrates family resilience” as “circumstances in which some family symptoms exist in the family and family functioning is diminished” based on Hohashi’s definition of family resilience [1]. “Family symptoms” are “problematic conditions (such as problems, issues, difficulties, or suffering) in the family system unit, comprehensively assessed by a nursing professional based on subjective and objective family data” [20]. “Family functions” are defined as “cognitive functions of the family and its cognitive power over the family and the family environment, which arise from the family role actions conducted by individual family members” [1]. The family system unit is “another term for family to clarify that the family is a system and a unit” [1].

In the CSFET (Figure 1) [18], “structural distance” means “the existence of family environment from the physical/objective perspective, and the degree of physical/objective separation from the family system unit”. “Functional distance” means “the condition of family environment from the psychological/subjective perspective, and the degree of psychological/subjective separation from the family system unit”. “Temporal distance” means “the transformation and processes thereof of family system unit and family environment from the chronological perspective, and the duration of separation between that particular moment and the present”. In addition, “supra-system” means “the outer frame that creates the family environment system, which is directly or indirectly related to other family environment systems and encompasses the family environment in its entirety”. “Macrosystem” means “family members’ sphere of daily activities that is distant from the family system unit, based on comprehensive physical/objective and psychological/subjective assessments”. “Microsystem” means “a familiar area in the neighborhood of the family system unit, based on comprehensive physical/objective and psychological/subjective assessments”. “Chronosystem” means “a concept to indicate the process of temporal change and transformation of the family internal environment system, family external environment system, and family system unit in a timeframe from the past to the future”.

### 2.2. Literature Collections

We carried out searches on Ichushi-Web, Japan’s only comprehensive medical literature database limited to studies in Japan, and we set the target year to “all”, and the article type to “excluding conference proceedings”. The search keywords were “family AND resilience”, “family resilience”, “family AND resiliency”, “family resiliency”, and finally, all the OR searches.

The subject keywords yielded 350 articles up to December 2019 (Figure 2). With reference to the title and abstract, we checked that the content described a situation in which the family demonstrated family resilience as an inclusion criterion. Even when a search obtained a hit, if the literature did not necessarily focus on “family resilience” but rather mostly discussed “individual resilience”, it was excluded. In addition, no restrictions were placed on demographic characteristics, and others. A total of 16 qualitative studies [21,22,23,24,25,26,27,28,29,30,31,32,33,34,35,36], including 22 family cases, were included in the study that provided examples of families undergoing difficulties due to some family symptoms and subsequently demonstrated family resilience (Table 1). The contents of the meaning units (labels) were extracted based on the similarities and differences according to the method of Elo and Kyngäs [37].

### 2.3. Family Interview

#### 2.3.1. Participants

The interviews were conducted in two areas in Japan from 2017 to 2019: an urban city with a population of approximately 1.5 million, and a rural city on a remote island with a population of approximately 34,000. Because families in their child-rearing years are more likely to experience adverse events [6], we targeted parenting families who were aware that family resilience is lacking or had been lacking. Family resilience is a concept that is self-perceived, and we thought that families that perceive that they lack family resilience would be more likely to discuss their family resilience situations. Participants were recruited through distribution research requests, clarifying the purpose of inquiring about family conditions demonstrating family resilience. In the urban city, 1820 copies were distributed (427 copies in 2017 and 1393 copies in 2018) to 17 facilities, including pediatric outpatient clinics, nursery schools, and so on. In the rural city, 545 copies were distributed to 14 facilities in 2019, including a kindergarten, a special support school, and so on. Thirty-four families applied (seven in 2017, eighteen in 2018, and nine in 2019), and we explained the outline of the study a second time. In this study, the target families were those that agreed to participate, and no criteria for inclusion or exclusion were set for families. We obtained letters of consent from 28 families and conducted interviews with mothers and/or fathers (5 in 2017, 14 in 2018, and 9 in 2019; Table 2). We continued recruiting families for 3 years from 2017 to 2019, and determined that saturation was reached in 2019, resulting in interviews with a total of 28 families.

#### 2.3.2. Data Collection

The data were collected through semi-structured interviews based on the Family Environment Observation/Interview (FEO/I) [19], Family Environment Map (FEM) [38], and Family Environment Assessment Index (FEAI) [39]. Because this study is based on the CSFET, all tools utilized were developed by Hohashi based on the CSFET.

We first used FEM, an assessment tool to understand the composition of the family and the relationship within and outside the family. The FEM is a qualitative tool for visualizing the family structure and kin structure and relationships between individuals. The interview guide was constructed based on the FEAI. The FEAI is a qualitative tool for assessing the family well-being and problematic family conditions. It is composed of 43 items and is provided with sample questions to conduct the culturally congruent family assessments. Examples of questions are: What sort of situation has your family found to be difficult? How did your family handle the situation? The FEO/I is “a format for centralizing and recording various information required for family assessments, family interviews/meetings, and so on”, and we always used the FEO/I during the interview period.

The interviews were recorded using an IC recorder with the consent of the participants, and a verbatim record was created. The average interview length was 83 min (range: 50–118 min).

### 2.4. Data Analysis and Ensuring Rigor

The two data sources, contents of the literature and semi-structured interviews, were used to enhance the trustworthiness of the analysis using the triangulation method. In order to clarify the state of the family in which family resilience was demonstrated, directed content analysis [40] was conducted based on the CSFET. The goal of a directed content analysis approach is to validate or extend conceptually a theoretical framework or theory. Existing theory or research can help focus the research question and it can help researchers begin by identifying key concepts or variables as initial coding categories [40].

The contents of the literature and family interview transcripts were combined, and the nine researchers independently read them numerous times to achieve a correct understanding. Sentences describing the state of family resilience were extracted and summarized to create a single meaning unit. If any disagreements arose among the nine researchers, the contents of the literature and family interview transcripts were re-read and re-evaluated. If no agreement was reached between the researchers, a third party consisting of 13 researchers exchanged opinions and determined the final evaluation. We also included situations in which a family overcame difficulties due to the effects of a family member’s behavior on the entire family. After summarizing the meaning units into subcategories and categories according to the commonality of contents, they were classified according to the perspective of four systems of the CSFET: family external environment system, family internal environment system, family system unit, and family chrono environment system. When classification was unclear, we returned to the definition of the CSFET theory and reconsidered classification. Nine researchers with expertise in the field of family nursing were involved in the analysis process until a consensus was reached. In addition, we received repeated opinions from 13 researchers to ensure the trustworthiness of the analysis. The results of the analysis were sent by mail to the 28 families that were interviewed to confirm that the content contained no discrepancies.

### 2.5. Ethical Considerations

This study was reviewed and approved by the institutional review board of the researchers’ university (Reference No. 454-3). When family members took part in semi-structured interviews, both orally and in writing, they received an explanation of the purpose of the study, anticipated risks and benefits, confidentiality protection, and the participants’ right to refuse participation or to withdraw from the research at any time.

## 3. Results

A total of 157 labels were extracted and, for a conceptual approximation to categories, were summarized in 47 subcategories and in 23 categories, and were placed in 4 themes (Table 3).

### 3.1. Theme 1: Family External Environmental System

#### 3.1.1. Can Utilize Relatives

Families overcame family difficulties by utilizing their relatives. Family members required hospitalization or care due to illness, disability, becoming pregnant, death, or when multiples of these family events occurred. When such events made it difficult for them to live on their own, the family sought support from the couple’s parents or siblings to help with household chores and childcare, and in some cases, to have them come to stay with them at home. Family also used their relatives not only for direct support, but also for psychological support. For example, they shared concerns about family difficulties with relatives.

#### 3.1.2. Can Utilize Friends

Families overcame family difficulties by utilizing friends. Family members were able to discuss their internal family problems with close friends on a regular basis, which spared them from having to deal with family problems and helped them to overcome family difficulties. By actively building friendships at school, they were able to have a community outside of the family, which provided psychological support to family members and helped them to overcome family difficulties.

#### 3.1.3. Can Utilize Co-Workers

Families overcame family difficulties by utilizing co-workers. Speaking to the co-workers about family concerns, such as the illness and death of a family member or the raising of a disabled child, helped family members to avoid keeping family problems within the family and to overcome difficulties. In addition, going to the workplace and getting involved with co-workers gave the family members a chance to interact with people outside the family, which helped family members feel relaxed and relieved, and helped family members to overcome their difficulties.

#### 3.1.4. Can Utilize Peers

Families could overcome difficulties by utilizing someone who had experienced a similar event. If a family hardship arose due to an event that was not previously experienced, the family sought solutions by approaching others to hear about their experiences of how they actually dealt with the event and asking them for advice on their current family situation. They also interacted with mothers of children with the same disabilities as their own children and attended family group sessions for psychological support.

#### 3.1.5. Can Utilize People in Locality

Families were able to overcome family difficulties by utilizing people in their locality. Families of children with disabilities were able to overcome difficulties during disasters and other emergencies because they formed a supportive relationship with people in the community who were able to help them on a regular basis. They also used the local community for psychological support by using it as a refuge from family difficulties.

#### 3.1.6. Can Utilize Professionals

Families overcame family difficulties by utilizing professionals. When a family member had an illness or disability, the family member shared concerns with a physician, nursing professional, or counselor for psychological support. Families were also able to obtain the professional support they needed for their families and family members by seeking support from professionals.

#### 3.1.7. Can Utilize School

Families can use the school to help them overcome their difficulties. Talking to the school about their child’s disability helped them obtain psychological support, and discussing their child’s school life helped them to create a suitable educational environment for their child.

#### 3.1.8. Can Utilize Social Welfare System

Families overcame family difficulties by utilizing the social welfare system. In the event of a family member’s death, they were able to provide financial support by using survivor’s pensions and the social welfare system for disabled children. In addition, using the facilities for people with disabilities also allowed them to interact with society.

#### 3.1.9. Can Obtain Support from Religion

Families overcame family difficulties by obtaining support from religion. During a period of time that families were debilitated by the stress of parenting, the family relied on the inspiration they received from their original religion for spiritual support.

### 3.2. Theme 2: Family Internal Environmental System

#### 3.2.1. Family Members Can Communicate with Members of Other Families

In families that overcame difficulties, family members communicated with members of other families to help the family overcome its difficulties. Family members shared their concerns with members of other families about problems that occurred within the family, problems between family members, and family members’ own problems. Family members were also able to alleviate their own psychological stress by sharing their concerns about their children’s disabilities and childcare with members of other families, rather than keeping them to themselves. In addition, when a family member was in distress, the family member’s suffering was alleviated when members of other families approached the family members to acknowledge their pain and feelings and to offer beneficial advice. Discussions among family members about family problems led to practical solutions and helped the family to overcome difficulties.

#### 3.2.2. Family Members Can Share Information

In families that overcame difficulties, family members shared information with each other to help the family overcome its difficulties. After receiving the diagnosis, family members had acquired the correct knowledge of their child’s disability by researching books, the internet, and by inquiring to experts. The gathering of information about treatment and counseling also helped the family to provide appropriate modifications for family members with disabilities. Sharing what one family member knew about the care and education of children with disabilities with other family members helped family members to accept the family’s current situation.

#### 3.2.3. Family Members Can Accept Current Situation

In families that overcame difficulties, family members accepted the current situation to help the family overcome their difficulties. Family members adapted to the situation by understanding the current situation of different values and the difficulties arising from other family members’ disabilities. In addition, family members were able to change the situation by reviewing their own behavior through objectively examining the family member’s situation as well as that of the family as a whole. Families with children having allergies and disabilities were able to switch their minds and change the way they perceive things by taking a positive view of the current situation instead of being pessimistic. Family members who were initially unable to accept the family member’s disability were able to accept the disability by expressing their thoughts and feelings about the disability, which ultimately helped the family member to overcome difficulties.

#### 3.2.4. Family Members Can Clearly Understand Their Roles

In families that overcame difficulties, family members clearly understood their roles to help the family overcome their difficulties. There was a shift in family member roles, with the grandmother taking the mother’s place in raising the child and fulfilling the mother’s role in the case of a family with a neglectful mother, and the grandfather fulfilling the father’s role in the case of a family where the father was not present owing to a single mother household. There was also a subjective family belief that if the father himself could recognize his role as a father and what he should do for his family as a father, they could overcome their current family difficulties. In addition, the division of household and childcare roles among family members reduced the members’ respective burdens.

#### 3.2.5. Family Members Can Support Other Family Members

In families that overcame difficulties, family members supported other family members to help the family overcome their difficulties. The family began to regain its balance, as other family members encouraged the independence of family members who had been confined to their rooms or who had misbehaved. Also, when a family member became fatigued or neurotic due to the family member’s childcare, disability, or household chores, or when a family member’s disability necessitated special accommodations, family members acted on behalf of other family members by helping with household chores, childcare, and helping with mood swings.

#### 3.2.6. Appropriate Distance Can Be Maintained Between Family Members

Families overcame family difficulties by maintaining an appropriate distance between family members. The panic-prone members of the family dealt with each other by keeping a certain distance between them so that each of them did not panic. Taking care not to involve the children in parental issues helped a family to get through a divorce, which can be considered a difficult family affair.

#### 3.2.7. Can Maintain Family Members’ Beliefs for Emotional Support

Families overcame family difficulties by holding beliefs for supporting family members. Families that did not have good marital relationships were able to raise their children by supporting the family’s previously held beliefs.

### 3.3. Theme 3: Family System Unit

#### 3.3.1. All Family Members Can Communicate with One Another

In families that overcame difficulties, all family members communicated with one another to help the family overcome its difficulties. Family members shared their concerns within the family by opening up about what they found difficult about raising a child with a disability. Also, when a family member was unaided in caring for a child with a disability, the family communicated with one another and worked together within the family.

#### 3.3.2. All Family Members Can Share Information

In families that overcame difficulties, all family members shared information to help the family overcome its difficulties. Families with low-birth-weight babies were able to share all aspects of information known by family members about their disability with all family members, and significant differences in the amount of information within the family were not observed.

#### 3.3.3. All Family Members Can Understand the Current Situation

In families that overcame difficulties, all family members understood the current situation to help the family overcome its difficulties. When a family member had a problem, such as caregiving, inheritance, or a scandal caused by a family member, or when a child’s university education became a financial hardship due to increased expenses, the family was able to deal with the situation constructively by understanding what was occurring. The family also properly identified what was currently needed for the family member with a disability and took the necessary actions.

#### 3.3.4. All Family Members Can Cooperate with One Another

In families that overcame difficulties, all family members cooperated with one another to help the family overcome its difficulties. All family members worked together and helped each other to deal with the complexities of international marriage, litigation involving family members who died as a result of a mishap, and family financial difficulties.

### 3.4. Theme 4: Family Chrono Environment System

#### 3.4.1. Can Utilize Past Experiences

Families overcame family difficulties by utilizing past experiences. The family drew on the family members’ childcare experience and professional experiences to raise their disabled children. The family was also confident in the fact that the family had overcome many difficulties in the past and was dealing with new challenges.

#### 3.4.2. Can Share Time with Family

Families overcame family difficulties by sharing time with family. Spending time together with other family members or with the family as a whole made a positive difference in family relationships. Children who had been institutionalized because of difficulties with family members’ relationships with each other formed good family relationships by being trained how to make their way home, and families that had difficulty accepting their children with disabilities became more accepting of their disabilities and felt like a family when they spent time together and, as a result, awareness of being a family began to grow. In addition, relationships within the family improved as family members quit their jobs to spend more time with their families.

#### 3.4.3. Can Share Objectives with Family

Families overcame family difficulties by sharing objectives with the family. Families whose children were in the Neonatal Intensive Care Unit (NICU) could target other children who were being discharged ahead of them, and families whose children had disabilities could overcome the difficulties that were currently occurring by targeting other children with the same disabilities.

## 4. Discussion

Categories were classified according to the system of the CSFET (Figure 1): these included nine categories for the family external environment system, seven categories for the family internal environment system, four categories for the family system unit, and three categories for the family chrono environment system.

In the family internal environment system, it was found that the conditions were generally easier to visualize, such as family cooperation and communication between family members to overcome family difficulties. On the other hand, the condition “family members can maintain an appropriate distance between family members” indicates that families need distance and that spending time together does not necessarily lead to overcoming difficulties. These considerations suggest that support for the family internal environment system should not only foster engagement between family members but also support distance between them [41].

As Walsh [17] pointed out, healthy families have the strength to admit when they need help and are more likely to turn to extended family, friends, neighbors, community services, or counseling. That is, not only the internal relations within the family but also the relations with the outside of the family help the family to overcome difficulties. In the family external environment system, it was indicated not only to be a condition of seeking support from outside the family, such as through “can ask relatives for support”, “can ask peers for advice”, and “can seek professional for help”, but also a condition of being able to change one’s mood by interacting with people outside the family, such as “can interact with friends”, “can interact with people at workplace”, “can interact with peers”, and “can interact with people in locality”, which can lead to positive results. In addition, it was found that family members felt a sense of relief when they shared their concerns about family difficulties with people outside the family, such as “can share concerns with relatives”, “can share concerns with friends”, “can share concerns with people at workplace”, “can share concerns with professionals”, and “can share concerns with school”, rather than through substantial cooperation, and that people outside the family were able to provide emotional support to the family members. The findings also indicated religion-related conditions, such as “can obtain support from religion”, and spirituality-related conditions, such as “can maintain family members’ beliefs for emotional support”, that were related to one’s beliefs. These were also evident in a study that conducted a conceptual analysis of family resilience and identified the factors that constitute family resilience, which stated that spirituality and religious beliefs appear to serve as powerful protectors often embraced by resilient families [13].

Furthermore, from the perspective of the family chrono environment system, which is a major asset of CSFET, it is clear that family resilience is related to the family chronicles in family nursing. We believe that the subcategories cover a wide range of family resilience, from family internal environmental systems to family chrono environmental systems [42].

The family conditions for family resilience revealed in this study may lead to more comprehensive understanding through consideration of the cultural and social context of Japan. The formation of resilience in Japanese families may involve unique social and family dynamics, such as traditional views of family, Confucian values, mutual assistance between generations, and community spirit in the local community [43]. For example, Confucian values are deeply rooted in Japanese society and views of family, and when faced with difficulties, the tendency to emphasize mutual assistance within the family and intergenerational connections may support family resilience. In future research, it will be important to clarify the uniqueness and universality of Japanese family resilience through comparisons with other cultures.

Families lacking in family resilience are unable to overcome difficult situations on their own and, therefore, need family nursing intervention to draw on their strengths. Nursing professionals have a major role to play in helping patients’ families overcome difficult situations on their own. However, because it is unclear which family members warrant intervention, and as it may be difficult for families to request assistance from a third party, families lacking family resilience are often unable to obtain support from a third party when encountering troublesome situations and are, therefore, unable to overcome their difficulties [44]. We believe that with such families, an assessment scale for family resilience, to be developed based on this study, will be useful for nursing professionals to identify families that are lacking in family resilience and who need intervention, and in assisting them in receiving support from a third party. If this scale can be developed, it will be easier for families with low family resilience to obtain support from third parties. In addition, this study will enable holistic assessment of family resilience, not only of the family system unit based on CSFET, but also of the family internal environment system, family external environment system, and family chrono environment system, which will be linked to support. Family conditions that demonstrate family resilience were also identified in this study. When a family confronts problematic conditions (such as problems, issues, difficulties, or suffering), nurses should consider the categories in this study when advising the family on what actions they should take and providing family intervention.

Japanese has no word that corresponds to resilience, so it is often written as-is, as a foreign word. Our future goal will be to assemble a wider range of literature by expanding the search to include “resilient”, “recover”, “rebound”, and other related terms. Because the literature used in this study was limited to studies in Japan, inclusion of studies outside Japan would also be desirable. In particular, from the perspective of CSFET, differences exist in family beliefs and economic power in the family system unit, as well as in the religion and culture in the family external environment system, so the results of the study are likely to vary according to the country or region. This study targeted families in the parenting period. Although data on families were refined, because the characteristics of families in the child-rearing phase are thought to have an impact on the results, there is a need to increase the number of attributes of the target families and continue to conduct semi-structured interviews to further refine the data and generalize the results. Examples of target families for which new data need to be added in the future include families with a family member in need of care, families with a family member with a disability, irrespective of the child-rearing period, and older adult-only households. In addition, the literature used in this study was up to 2019. Since then, there may have been research on family resilience, particularly in light of the impact of COVID-19. Future research that incorporates new literature is expected. Note that the purpose of this study was to clarify family conditions demonstrating family resilience. The problems, issues, difficulties, or suffering that cause a decline in family resilience, or what resilience factors contribute to improving family functioning, were not researched; it will, however, be important to clarify these.

## 5. Conclusions

The concept of family resilience has been gaining attention in family nursing. Hohashi defined family resilience as “when a family becomes aware of family symptoms, its power to autonomously and actively improve its own family functions”. This study used literature and semi-structured interviews to identify conditions that demonstrate family resilience. A total of 157 labels, 47 subcategories, and 23 categories were extracted. The categories were classified according to the system of Concentric Sphere Family Environment Theory (CSFET): nine categories for the family external environment system, seven categories for the family internal environment system, four categories for the family system unit, and three categories for the family chrono environment system. A variety of states were analyzed, such as family member interactions, whole family interactions, use of social resources, and religious and spiritual support, revealing the diversity of family resilience.

## Figures and Tables

**Figure 1 nursrep-15-00096-f001:**
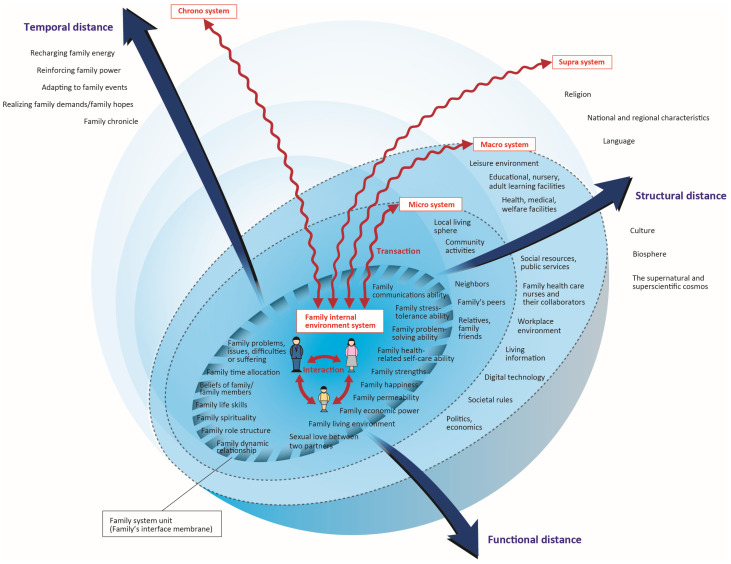
Model diagram of Concentric Sphere Family Environment Theory (CSFET; ver. 3.4).

**Figure 2 nursrep-15-00096-f002:**
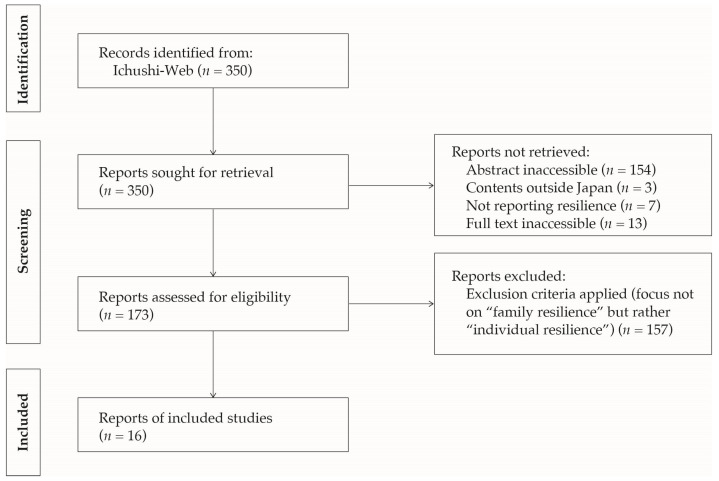
The flow of study identification and selection.

**Table 1 nursrep-15-00096-t001:** Sources of family cases in the literature (*N* of references = 16; *N* of family cases = 22).

Case Number	Family Case	Reference
1	A family having a child with nervous tic symptoms	[21]
2	A family whose eldest son had schizophrenia and was confined to his room	[22]
3	A family whose children had been left with severe disabilities resulting from an accident	[23]
4	A family whose eldest son was a homicide victim	[24]
5	A family in which a difference in values emerged between the husband and wife	[25]
6, 7	A family that had difficulty accepting their child’s disability	[26]
8	A family with a child in need of medical care	[27]
9	A family that kept their disabled daughter locked up in their home for decades	[28]
10	A family of a single mother raising a developmentally disabled child	[29]
11	A family in which the parents are divorced and the father and daughter are living together	[30]
12	Families in which the father physically abuses the daughter, the daughter behaves delinquently, and the daughter and stepmother do not get along	[31]
13	A family whose father was hospitalized for subarachnoid hemorrhage	[32]
14	A family of a child with disabilities that was affected by the disaster	[33]
15	A family in which the husband and the husband’s mother are cancer patients and the husband is terminally ill	[34]
16, 17, 18, 19, 20, 21	A family whose children were born with low birth weight	[35]
22	A family with a dysfunctional marriage, a mother with abusive experiences and depression, and a child with developmental disabilities	[36]

**Table 2 nursrep-15-00096-t002:** Sociodemographic characteristics of families participating in the semi-structured interviews (*N* of families = 28).

Characteristics	M	SD	Range
Age (years)			
Husband	40.18	5.37	30–49
Wife	38.68	5.47	27–49
*N* of family members	8.14	4.36	3–18
*N* of children	2.29	1.01	1–5
*N* of family member(s) with a disease/illness/disability	2.07	1.49	0–6
*N* of husband’s sibling(s)	2.43	1.21	1–5
*N* of wife’s sibling(s)	3.14	2.01	1–12
	** *n* **	**%**	
Family type			
Nuclear family	26	92.9	
Extended family	2	7.1	
Family structure			
Two-parent family	26	92.9	
Single mother family	2	7.1	
Husband employed ^1^			
Yes	26	100	
No	0	0	
Wife employed			
Yes	20	71.4	
No	8	28.6	

Note: ^1^
*N* of families with a husband was 26 in total.

**Table 3 nursrep-15-00096-t003:** Identified family conditions for family resilience (literature: *N* of cases = 22; family interviews: *N* of families = 28).

System	Category	Subcategory
Family external environment system	Can utilize relatives	Can share concerns with relatives Can ask relatives for support
	Can utilize friends	Can share concerns with friends Can interact with friends
	Can utilize co-workers	Can share concerns with people at workplace Can interact with people at workplace
	Can utilize peers	Can interact with peers Can listen to peers Can ask peers for advice Can take part in the family group
	Can utilize people in locality	Can establish a supportive relationship with people in locality Can interact with people in locality
	Can utilize professionals	Can share concerns with professionals Can seek professional for help
	Can utilize school	Can share concerns with school Can discuss matters with school
	Can utilize social welfare system	Can make use of social insurance system Can make use of social welfare system
	Can obtain support from religion	Can obtain support from religion
Family internal environment system	Family members can communicate with members of other families	Family members can share concerns with other family members Family members can understand feelings of other family members Family members can advise other family members Family members can talk to other family members
	Family members can share information	Family members can share information that family needs Family members can gather information
	Family members can accept current situation	Family members can accept current situation of other family members Family members can accept current situation of the family Family members can accept current situation positively Family members can accept the disability of other family members
	Family members can clearly understand their roles	Family can make the family role transition Family members can recognize their own family members’ roles Family members can share the family role
	Family members can support other family members	Can encourage the autonomy of family members Family members can act to support other family members
	Appropriate distance can be maintained between family members	Family members can maintain an appropriate distance between family members Can keep other family members out of trouble
	Can maintain family members’ beliefs for emotional support	Can maintain family members’ beliefs for emotional support
Family system unit	All family members can communicate with one another	All family members can share concerns with one another All family members can talk to one another
	All family members can share information	All family members can share information
	All family members can understand the current situation	All family members can understand the current situation
	All family members can cooperate with one another	All family members can cooperate with one another
Family chrono environment system	Can utilize past experiences	Family members can utilize past experiences Family can utilize past experiences
	Can share time with family	Family members can share time with family All family members can share time with family
	Can share objectives with family	Can share objectives with family

Note: System classifications are based on systems of Concentric Sphere Family Environment Theory (CSFET).

## Data Availability

The data are not publicly available for reasons of privacy or due to ethical restrictions.

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
