# Peer review of "Japanese Family Conditions Demonstrating Family Resilience: Directed Content Analysis Based on Literature and Family Interviews"

_nursrep, 2025, doi:10.3390/nursrep15030096_

Round 1
Reviewer 1 Report
Comments and Suggestions for Authors
Thank you for inviting me to review the manuscript, “The family conditions demonstrating family resilience: Directed content analysis based on literature and family interviews.” Overall, this is a well-written and well-structured manuscript that builds on solid theoretical and empirical foundations.
However, I believe the manuscript would benefit from additional clarification and elaboration in several key areas. Below, I outline my comments and suggestions for improvement:
In the introduction (Lines 33–34), the statement that “In today’s society, … family resilience is declining, making it more difficult for families to solve problems on their own” may not universally hold true and may be highly context-dependent. For instance, in individualistic societies, families may face challenges such as isolation or weakened community ties but might also have access to professional resources and social policies that foster resilience. Additionally, the manuscript does not provide a definition of "family resilience" before this statement is introduced, which could help readers contextualize the argument.
The focus on Japanese families is commendable; however, this cultural specificity may limit the generalizability of findings to other societies. It would be helpful to discuss how these findings apply to broader cultural contexts or, alternatively, to clarify this cultural scope in the title (e.g., explicitly mentioning "Japanese families").
The manuscript mentions using terms like “family AND resilience,” “family resilience,” “family AND resiliency,” and “family resiliency.” However, these terms may not capture the breadth of related literature. Studies used other related terms such as “resilient,” “recover*,” and “rebounc*” might have been missed. This needs to be acknowledged as a limitation.
The literature search is limited to publications up to December 2019. Given the significant developments in family resilience research, particularly post-COVID-19, which has been a major stressor for families globally, updating the search to include studies through December 2024 would add critical value to this work.
Including a PRISMA flow diagram to illustrate the search, screening, and selection process, along with inclusion and exclusion criteria, would enhance transparency and methodological rigor. For reference, see the PRISMA 2020 guidelines https://www.prisma-statement.org/prisma-2020-flow-diagram
The procedure of recruiting families for family interviews is not clear, such as How were families contacted? How many families were contacted, and how many agreed to participate? What were the inclusion and exclusion criteria for participant selection? How was the sample size determined?
The manuscript would benefit from a clearer explanation of how findings from the literature review and family interviews were integrated. Did the literature review predefine themes that were then explored in interviews? What unique insights were gained from the interviews that extend beyond the literature review?
While the manuscript focuses on Japanese families, more discussion is needed on the cultural context of family resilience in Japan and how these findings relate to or differ from resilience in other cultures. For example, are there specific societal, structural, or familial dynamics unique to Japanese families that shape resilience?
Author Response
Comments 1: Thank you for inviting me to review the manuscript, “The family conditions demonstrating family resilience: Directed content analysis based on literature and family interviews.” Overall, this is a well-written and well-structured manuscript that builds on solid theoretical and empirical foundations.
However, I believe the manuscript would benefit from additional clarification and elaboration in several key areas. Below, I outline my comments and suggestions for improvement:
In the introduction (Lines 33–34), the statement that “In today’s society, … family resilience is declining, making it more difficult for families to solve problems on their own” may not universally hold true and may be highly context-dependent. For instance, in individualistic societies, families may face challenges such as isolation or weakened community ties but might also have access to professional resources and social policies that foster resilience. Additionally, the manuscript does not provide a definition of "family resilience" before this statement is introduced, which could help readers contextualize the argument.
Response 1: The following revisions were made in the paragraph from line 32.
Family nursing aims at maintaining and improving family functioning in the target family system unit, and it targets the family as a living organization. However, in today's Japan, families are becoming more diverse, family ties are weakening, and family functioning has been declining, making it more difficult for families to solve problems on their own [1]. It is considered that rather than unilateral family intervention/support from others to solve problems, families are more likely to improve their self-care skills through the support of others and to continue to practice the solutions they themselves have devised, which are more likely to be put into practice [1]. Under these circumstances, the concept of "family resilience" has attracted much attention in family nursing.
Comments 2: The focus on Japanese families is commendable; however, this cultural specificity may limit the generalizability of findings to other societies. It would be helpful to discuss how these findings apply to broader cultural contexts or, alternatively, to clarify this cultural scope in the title (e.g., explicitly mentioning "Japanese families").
Response 2: As you suggested, the title of the manuscript has been changed to "Japanese family conditions demonstrating family resilience: Directed content analysis based on literature and family interviews."
Also, from line 478 onwards, we have added more detail to the limitation of the study, which was the need for research outside of Japan.
Because the literature used in this study was limited to studies in Japan, inclusion of studies outside Japan would also be desirable. In particular, from the perspective of CSFET, differences exist in family beliefs and economic power in the family system unit, as well as in the religion and culture in the family external environment system, so the results of the study are likely to vary according to the country or region.
Comments 3: The manuscript mentions using terms like “family AND resilience,” “family resilience,” “family AND resiliency,” and “family resiliency.” However, these terms may not capture the breadth of related literature. Studies used other related terms such as “resilient,” “recover*,” and “rebounc*” might have been missed. This needs to be acknowledged as a limitation.
Response 3: The following was added to line 476:
Japanese has no word that corresponds to resilience, so it is often written as-is, as a foreign word. Our future goal will be to assemble a wider range of literature by expanding the search to include "resilient," "recover," "rebound" and related terms.
Comments 4: The literature search is limited to publications up to December 2019. Given the significant developments in family resilience research, particularly post-COVID-19, which has been a major stressor for families globally, updating the search to include studies through December 2024 would add critical value to this work.
Response 4: As this is a long-term study, the literature included is from before the COVID-19 pandemic. The following is stated from line 490 onwards as one of the limitations of the study.
In addition, the literature used in this study is up to 2019. Since then, there may have been research on family resilience, particularly in light of the impact of COVID-19. Future research that incorporates new literature is expected.
Comments 5: Including a PRISMA flow diagram to illustrate the search, screening, and selection process, along with inclusion and exclusion criteria, would enhance transparency and methodological rigor. For reference, see the PRISMA 2020 guidelines https://www.prisma-statement.org/prisma-2020-flow-diagram
Response 5: On line 149, Figure 2 has been added.
Comments 6: The procedure of recruiting families for family interviews is not clear, such as How were families contacted? How many families were contacted, and how many agreed to participate? What were the inclusion and exclusion criteria for participant selection? How was the sample size determined?
Response 6: Your question is explained on lines 155 to 172.
The interviews were conducted in two areas in Japan from 2017 to 2019: an urban city with a population of approximately 1.5 million, and a rural city on a remote island with a population of approximately 34,000. Because families in their child-rearing years are more likely to experience adverse events [6], we targeted parenting families who were aware that family resilience is lacking or had been lacking. Family resilience is a concept that is self-perceived, and we thought that families that perceive that they lack family resilience would be more likely to discuss their family resilience situations. We recruited participants through distribution research requests. In the urban city, 1,820 copies were distributed (427 copies in 2017, 1,393 copies in 2018) to 17 facilities, including pediatric outpatient clinics, nursery schools, and so on. In the rural city, 545 copies were distributed to 14 facilities in 2019, including a kindergarten, a special support school, and so on. Thirty-four families applied (seven in 2017, 18 in 2018, nine in 2019), and we explained the outline of the study a second time. In this study, the target families were those that agreed to participate, and no criteria for inclusion or exclusion were set for families. We obtained letters of consent from 28 families, and conducted interviews with mothers and/or fathers (five in 2017, 14 in 2018, nine in 2019) (Table 2). We continued recruiting families for three years from 2017 to 2019, and determined that saturation was reached in 2019, resulting in interviews with a total of 28 families.
As no specific criteria for inclusion or exclusion were set, we added the following on line 167: In this study, the target families were those that agreed to participate, and no criteria for inclusion or exclusion were set for families.
As for sample size, we continued recruiting families until data saturation was reached, so the following was added on line 170:
We continued recruiting families for three years from 2017 to 2019, and determined that saturation was reached in 2019, resulting in interviews with a total of 28 families.
Comments 7: The manuscript would benefit from a clearer explanation of how findings from the literature review and family interviews were integrated. Did the literature review predefine themes that were then explored in interviews? What unique insights were gained from the interviews that extend beyond the literature review?
Response 7: This study is a triangulation of literature and family interview data. In other words, after combining the contents (meaning units) of the literature and family interviews, subcategories and categories were created. Details of the data analysis are explained on lines 196 to 220.
The two data sources, contents of literature and semi-structured interviews, were used to enhance the trustworthiness of the analysis using the triangulation method. In order to clarify the state of the family in which family resilience is demonstrated, directed content analysis [25] was conducted based on the CSFET. The goal of a directed content analysis approach is to validate or extend conceptually a theoretical framework or theory. Existing theory or research can help focus the research question and it can help researchers begin by identifying key concepts or variables as initial coding categories [25].
The contents of literature and family interview transcripts were combined, and the nine researchers independently read them numerous times to achieve a correct understanding. Sentences describing the state of family resilience were extracted and summarized to create a single meaning unit. If any disagreements arose among the nine researchers, the contents of literature and family interview transcripts were re-read and re-evaluated. If no agreement was reached between the researchers, a third party consisting of 13 researchers exchanged opinions and determined the final evaluation. We also included situations in which a family overcame difficulties due to the effects of a family member's behavior on the entire family. After summarizing the meaning units into subcategories and categories according to the commonality of contents, they were classified according to the perspective of four systems of the CSFET: family external environment system, family internal environment system, family system unit, and family chrono environment system. When classification was unclear, we returned to the definition of the CSFET theory and reconsidered classification. Nine researchers with expertise in the field of family nursing were involved in the analysis process until a consensus was reached. In addition, we received repeated opinions from 13 researchers to ensure the trustworthiness of the analysis. The results of the analysis were sent by mail to the 28 families that were interviewed to confirm that the content contained no discrepancies.
Comments 8: While the manuscript focuses on Japanese families, more discussion is needed on the cultural context of family resilience in Japan and how these findings relate to or differ from resilience in other cultures. For example, are there specific societal, structural, or familial dynamics unique to Japanese families that shape resilience?
Response 8: The following has been added from line 446.
The family conditions for family resilience revealed in this study may lead to more comprehensive understanding through consideration of the cultural and social context of Japan. The formation of resilience in Japanese families may involve unique social and family dynamics such as traditional views of family, Confucian values, mutual assistance between generations, and community spirit in the local community [28]. For example, Confucian values ​​are deeply rooted in Japanese society and views of family, and when faced with difficulties the tendency to emphasize mutual assistance within the family and intergenerational connections may support family resilience. In future research, it will be important to clarify the uniqueness and universality of Japanese family resilience through comparisons with other cultures.
Reviewer 2 Report
Comments and Suggestions for Authors
Title: The family conditions demonstrating family resilience: Directed content analysis based on literature and family interviews
I found this to be a socially and scientifically very relevant paper. The authors in the study analyzed how the family conditions demonstrating family resilience. Family resilience is defined based on the Concentric Sphere Family Environment Theory (CSFET) as when a family becomes aware of family symptoms on its own, and its power to autonomously and actively improve its own family functions. The results of a literature review of 22 family nursing cases from 16 qualitative studies on family resilience and the results of 28 semi-structured interviews with 28 parenting-age families in Japan were qualitatively categorized using the triangulation method, and directed content analysis was conducted based on the CSFET. The authors discuss results in terms of a variety of diverse aspects contributing to a family’s resilience, including family member interactions, entire family interactions, use of social resources, and religious and spiritual support, were indicated.
The article has exactly clear aim which is fulfilled and the article grounded within relevant and current literature. The author/authors gives/give a review of contemporary research in the field. The article is well structured and clearly written, including introduction, objectives of the study, data and methods, analysis and results and discussion. The author/authors used appropriate methodology. The paper is written in a clear and concise style, and ides are well organized.
My suggestion for improving a quality of the paper and specific remarks have:
- in the introduction, add and define family resilience from some other well-known researchers in the field such as Masten et al.
- state what kind of possible implementation of the obtained results is
I recommend publishing the Article as original scientific paper with proposed changes.
Author Response
Comments 1: I found this to be a socially and scientifically very relevant paper. The authors in the study analyzed how the family conditions demonstrating family resilience. Family resilience is defined based on the Concentric Sphere Family Environment Theory (CSFET) as when a family becomes aware of family symptoms on its own, and its power to autonomously and actively improve its own family functions. The results of a literature review of 22 family nursing cases from 16 qualitative studies on family resilience and the results of 28 semi-structured interviews with 28 parenting-age families in Japan were qualitatively categorized using the triangulation method, and directed content analysis was conducted based on the CSFET. The authors discuss results in terms of a variety of diverse aspects contributing to a family’s resilience, including family member interactions, entire family interactions, use of social resources, and religious and spiritual support, were indicated.
The article has exactly clear aim which is fulfilled and the article grounded within relevant and current literature. The author/authors gives/give a review of contemporary research in the field. The article is well structured and clearly written, including introduction, objectives of the study, data and methods, analysis and results and discussion. The author/authors used appropriate methodology. The paper is written in a clear and concise style, and ides are well organized.
My suggestion for improving a quality of the paper and specific remarks have:
- in the introduction, add and define family resilience from some other well-known researchers in the field such as Masten et al.
Response 1: After the definition of family resilience by McCubbin and McCubbin [15], we added the definition of family resilience by Masten et al. from line 64 onwards.
McCubbin and McCubbin [15] have defined family resilience as "characteristics, dimensions, and properties of families which help families to be resilient to disruption in the face of change, and adaptive in the face of crisis situation" and Masten and Barnes [16] define family resilience as "the capacity of a family to handle a specific impending crisis", and the written meanings do not differ greatly depending on the researchers.
Comments 2: - state what kind of possible implementation of the obtained results is
I recommend publishing the Article as original scientific paper with proposed changes.
Response 2: The following was added to line 466.
If this scale can be developed, it will be easier for families with low family resilience to obtain support from third parties. In addition, this study will enable holistic assessment of family resilience, not only of the family system unit based on CSFET, but also of the family internal environment system, family external environment system, and family chrono environment system, which will be linked to support. Family conditions that demonstrate family resilience were also identified in this study. When a family confronts problematic conditions (such as problems, issues, difficulties or suffering), nurses should consider the categories in this study when advising the family on what actions they should take and providing family intervention.
Reviewer 3 Report
Comments and Suggestions for Authors
Dear Authors,
I appreciate the opportunity to review your manuscript. Your study presents valuable insights; however, I would like to offer suggestions that could enhance the quality of your paper.
Methods:
I address two key aspects of your methodology: the literature review approach and qualitative data analysis process.
First, the methodology section does not specify the type of literature review conducted. Identifying whether it is a systematic review, scoping review, integrative review, or another approach is essential to ensure transparency and methodological rigor. Additionally, outlining the criteria for selecting and analyzing sources would strengthen the reliability of your review. I recommend clarifying this aspect to provide a clearer understanding of how the literature was collected, evaluated, and synthesized.
Second, regarding qualitative data analysis, while the use of software such as Atlas.ti or NVivo is not mandatory, these tools enhance transparency, organization, and rigor. The absence of such software raises concerns regarding data management, coding consistency, and reproducibility. If a manual approach is used, I suggest providing a more detailed explanation of how coding was conducted, how themes were identified, and what measures were taken to ensure systematic and unbiased analysis.
To strengthen the methodological clarity and credibility of your study, I recommend expanding the methodology section to explicitly define the type of literature review conducted and to justify the chosen approach for data analysis.
Comments on the Quality of English Language
I recommend revising the english so that the language is more direct and appealing to readers.
Author Response
Comments 1: Dear Authors,
I appreciate the opportunity to review your manuscript. Your study presents valuable insights; however, I would like to offer suggestions that could enhance the quality of your paper.
Methods:
I address two key aspects of your methodology: the literature review approach and qualitative data analysis process.
First, the methodology section does not specify the type of literature review conducted. Identifying whether it is a systematic review, scoping review, integrative review, or another approach is essential to ensure transparency and methodological rigor. Additionally, outlining the criteria for selecting and analyzing sources would strengthen the reliability of your review. I recommend clarifying this aspect to provide a clearer understanding of how the literature was collected, evaluated, and synthesized.
Response 1: This study is not a literature review, nor is it a systematic review, scoping review, or integrative review. We searched for literature, extracted meaning units (labels) from the contents, combined them with the meaning units from the family interviews, triangulated, and then subcategorized and categorized. The subtitle on line 133 has been changed from "2.2. Literature Review" to "2.2. Literature Collections."
In addition, the following has been added from line 147.
The contents of the meaning units (labels) were extracted based on the similarities and differences according to the method of Elo and Kyngäs [22].
Comments 2: Second, regarding qualitative data analysis, while the use of software such as Atlas.ti or NVivo is not mandatory, these tools enhance transparency, organization, and rigor. The absence of such software raises concerns regarding data management, coding consistency, and reproducibility. If a manual approach is used, I suggest providing a more detailed explanation of how coding was conducted, how themes were identified, and what measures were taken to ensure systematic and unbiased analysis.
To strengthen the methodological clarity and credibility of your study, I recommend expanding the methodology section to explicitly define the type of literature review conducted and to justify the chosen approach for data analysis.
Response 2: This study is a triangulation of literature and family interview data, and the ensuring of rigor is described in lines 195 to 220. Its subtitle is "2.4. Data Analysis and Ensuring Rigor" and the rigor is explained in more detail.
The two data sources, contents of literature and semi-structured interviews, were used to enhance the trustworthiness of the analysis using the triangulation method. In order to clarify the state of the family in which family resilience is demonstrated, directed content analysis [25] was conducted based on the CSFET. The goal of a directed content analysis approach is to validate or extend conceptually a theoretical framework or theory. Existing theory or research can help focus the research question and it can help researchers begin by identifying key concepts or variables as initial coding categories [25].
The contents of literature and family interview transcripts were combined, and the nine researchers independently read them numerous times to achieve a correct understanding. Sentences describing the state of family resilience were extracted and summarized to create a single meaning unit. If any disagreements arose among the nine researchers, the contents of literature and family interview transcripts were re-read and re-evaluated. If no agreement was reached between the researchers, a third party consisting of 13 researchers exchanged opinions and determined the final evaluation. We also included situations in which a family overcame difficulties due to the effects of a family member's behavior on the entire family. After summarizing the meaning units into subcategories and categories according to the commonality of contents, they were classified according to the perspective of four systems of the CSFET: family external environment system, family internal environment system, family system unit, and family chrono environment system. When classification was unclear, we returned to the definition of the CSFET theory and reconsidered classification. Nine researchers with expertise in the field of family nursing were involved in the analysis process until a consensus was reached. In addition, we received repeated opinions from 13 researchers to ensure the trustworthiness of the analysis. The results of the analysis were sent by mail to the 28 families that were interviewed to confirm that the content contained no discrepancies.
Round 2
Reviewer 1 Report
Comments and Suggestions for Authors
Thank you for your responses to the previous review. I appreciate the revisions made so far.
However, I still have one key concern regarding the conceptualization of resilience in your study. Typically, resilience research considers adversity or stressful events as exposures, positive adaptation as the outcome, and protective factors as the mechanisms facilitating this process. In the current manuscript, it remains unclear how these components are integrated. Specifically, it would be helpful to clarify which family adversities or circumstances are linked to which positive outcomes, and through what resilience resources or factors. For instance, could Table 1 and Table 3 be better aligned to explicitly demonstrate the specific or common resilience factors associated with different types of family conditions?
Additionally, I noticed that no clear inclusion or exclusion criteria were set for family interviews. Without such criteria, it is unclear how family exposures were identified as triggers for the resilience process. Were there any questions about stress/exposures during the interviews? Perhaps presenting the types of exposures or family cases within Table 3 could help clarify this aspect.
Clarifying and discussing this point would be very helpful! Thank you!
Author Response
Comments 1: Thank you for your responses to the previous review. I appreciate the revisions made so far.
However, I still have one key concern regarding the conceptualization of resilience in your study. Typically, resilience research considers adversity or stressful events as exposures, positive adaptation as the outcome, and protective factors as the mechanisms facilitating this process. In the current manuscript, it remains unclear how these components are integrated. Specifically, it would be helpful to clarify which family adversities or circumstances are linked to which positive outcomes, and through what resilience resources or factors. For instance, could Table 1 and Table 3 be better aligned to explicitly demonstrate the specific or common resilience factors associated with different types of family conditions?
Response 1: The purpose of this study is to clarify family conditions demonstrating family resilience. In addition, based on the results of this study, a scale will be developed to screen for the degree of family resilience in the future. The definition of family resilience in this study appears in lines 84 to 86.
Family conditions are defined in lines 101 to 108. This study did not research what problems, issues, difficulties or suffering cause family conditions. Therefore, the following sentence has been added from line 494 as a future task.
Note that the purpose of this study is to clarify family conditions demonstrating family resilience. The problems, issues, difficulties or suffering that cause a decline in family resilience, or what resilience factors contribute to improving family functioning, were not researched; it will, however, be important to clarify these.
Comments 2: Additionally, I noticed that no clear inclusion or exclusion criteria were set for family interviews. Without such criteria, it is unclear how family exposures were identified as triggers for the resilience process. Were there any questions about stress/exposures during the interviews? Perhaps presenting the types of exposures or family cases within Table 3 could help clarify this aspect.
Clarifying and discussing this point would be very helpful! Thank you!
Response 2: As mentioned above, the purpose of this study is to clarify family conditions demonstrating family resilience, and research concerning what problems, issues, difficulties or suffering that were the cause of family conditions was not conducted.
Since multiple problems, issues, difficulties or suffering make up the family conditions of each family, no one-to-one correspondence exists between family conditions and problems, issues, difficulties or suffering.
Note that Table 3 is the result of data analysis of literature and family interviews, and it is difficult to list types of exposures here.
As appears in lines 157 to 159, we considered that families in their child-rearing years are more likely to experience adverse events, and targeted such families.
The purpose of the study was explained in the written recruitment documents for families, so the families participated in the interviews on the premise that they would be interviewed about family conditions demonstrating family resilience. Regarding this point, the following was added to line 161:
Participants were recruited through distribution research requests, clarifying the purpose of inquiring about family conditions demonstrating family resilience.
Reviewer 3 Report
Comments and Suggestions for Authors
No comments.
Author Response
Thank you for the peer review.